# Global Distribution of Canine Visceral Leishmaniasis and the Role of the Dog in the Epidemiology of the Disease

**DOI:** 10.3390/pathogens13060455

**Published:** 2024-05-27

**Authors:** Diego Fernandes Vilas-Boas, Eiji Kevin Nakasone Nakasone, Ana Alice Maia Gonçalves, Daniel Ferreira Lair, Diana Souza de Oliveira, Diogo Fonseca Soares Pereira, Geralda Gabriele Silva, Ingrid dos Santos Soares Conrado, Lucilene Aparecida Resende, Maykelin Fuentes Zaldívar, Reysla Maria da Silveira Mariano, Walderez Ornelas Dutra, Miguel Angel Chávez-Fumagalli, Alexsandro Sobreira Galdino, Denise Silveira-Lemos, Rodolfo Cordeiro Giunchetti

**Affiliations:** 1Laboratory of Biology of Cell Interactions, Institute of Biological Sciences, Department of Morphology, Federal University of Minas Gerais, Belo Horizonte 31270-901, MG, Brazil; diego.fervbio@gmail.com (D.F.V.-B.); eijitus@gmail.com (E.K.N.N.); anafish@hotmail.com (A.A.M.G.); daniel.lair@outlook.com (D.F.L.); diana.morgana@hotmail.com (D.S.d.O.); diogofsp@yahoo.com.br (D.F.S.P.); geraldagabrielevet@gmail.com (G.G.S.); ingrid_1709@yahoo.com.br (I.d.S.S.C.); lucilenearesende@yahoo.com.br (L.A.R.); maykelin.fuentes@gmail.com (M.F.Z.); reyslamariano@yahoo.com.br (R.M.d.S.M.); waldutra@icb.ufmg.br (W.O.D.); denise.lemos@gmail.com (D.S.-L.); 2Computational Biology and Chemistry Research Group, Vicerrectorado de Investigación, Universidad Católica de Santa María, Arequipa 04000, Peru; mchavezf@ucsm.edu.pe; 3Microorganism Biotechnology Laboratory, Federal University of São João Del-Rei (UFSJ), Midwest Campus, Divinópolis 35501-296, MG, Brazil; asgaldino@ufsj.edu.br

**Keywords:** *Leishmania infantum*, epidemiology, canine visceral leishmaniasis

## Abstract

Visceral leishmaniasis is a disease caused by protozoa of the species *Leishmania (Leishmania) infantum* (syn = *Leishmania chagasi*) and *Leishmania (Leishmania) donovani*, which are transmitted by hematophagous insects of the genera *Lutzomyia* and *Phlebotomus*. The domestic dog (*Canis familiaris*) is considered the main urban reservoir of the parasite due to the high parasite load on its skin, serving as a source of infection for sandfly vectors and, consequently, perpetuating the disease in the urban environment. Some factors are considered important in the perpetuation and spread of canine visceral leishmaniasis (CVL) in urban areas, such as stray dogs, with their errant behavior, and houses that have backyards with trees, shade, and organic materials, creating an attractive environment for sandfly vectors. CVL is found in approximately 50 countries, with the number of infected dogs reaching millions. However, due to the difficulty of controlling and diagnosing the disease, the number of infected animals could be even greater. In the four continents endemic for CVL, there are reports of disease expansion in endemic countries such as Brazil, Italy, Morocco, and Tunisia, as well as in areas where CVL is not endemic, for example, Uruguay. Socio-environmental factors, such as migration, drought, deforestation, and global warming, have been pointed out as reasons for the expansion into areas where it had been absent. Thus, the objective of this review is to address (i) the distribution of CVL in endemic areas, (ii) the role of the dog in the visceral leishmaniasis epidemiology and the factors that influence dog infection and the spread of the disease, and (iii) the challenges faced in the control of CVL.

## 1. Introduction

Leishmaniasis is a complex of diseases whose etiological agent is the digenetic protozoa (amastigote and promastigote forms) of the *Leishmania* genus. The transmission of the parasite occurs during the blood meal of female phlebotomine sandflies of the genus *Phlebotomus* in the Old World and *Lutzomyia* in the New World [1]. It is a disease considered neglected by the World Health Organization (WHO), with a wide geographic distribution and presence in 98 countries. Worldwide, it is estimated that there are about 12 million infected people, with approximately 700,000 to 1 million new cases reported each year [2,3,4,5]. Leishmaniasis has different clinical manifestations, which depend on the species of the parasite and the host’s immune response to the cutaneous, mucosal, and visceral forms [6].

Visceral leishmaniasis (VL) is the most severe form of the disease, with a high mortality rate in humans if left untreated [3]. It is estimated that 50,000 to 90,000 new cases of VL occur annually worldwide, with the majority occurring in Brazil and East African countries [5]. In the Americas, LV is present in more than 10 countries, such as Argentina, Bolivia, Brazil, Colombia, El Salvador, Guatemala, Honduras, Mexico, Paraguay, Uruguay, and Venezuela. In 2020, Brazil accounted for 97% (1933) of the total cases [7]. *Leishmania (Leishmania) donovani* is the species responsible for VL in Africa and Asia, while in the Mediterranean region and the Americas, the disease is caused mainly by *Leishmania (Leishmania) infantum* (syn = *Leishmania chagasi*) [6,8,9].

Many animals, such as armadillos, jackals, rodents, bats, cats, dogs, and mongooses, play an important role in the parasite’s life cycle, acting as hosts [3,10]. However, dogs are the main urban reservoirs of *L. infantum* in Brazil and several other countries [11,12]. In some regions, especially in South America, the seroprevalence can reach 33% on Margarita Island, Venezuela, and 36% in Jacobina, Brazil [13]. Canine visceral leishmaniasis (CVL) is present in approximately 50 countries; except for Antarctica and Oceania, all continents have endemic regions for the disease [14,15]. Dogs are often asymptomatic and, when symptomatic, can present a wide spectrum of clinical manifestations, such as anemia, emaciation, hepatosplenomegaly, renal changes, and onychogryphosis, which, if not treated, can lead to death [16]. Even when treated, the dogs remain infective to vectors, maintaining the presence of the disease in urban centers [17], and becoming a major public health problem.

Therefore, given the close relationship between the dog and the disease, as well as their strong interaction with humans, this review discusses the role of dogs in CVL epidemiology and the disease’s distribution in endemic areas, highlighting the main challenges in controlling CVL.

## 2. Distribution of Canine Visceral Leishmaniasis in the World

### 2.1. Africa

On the African continent, the most VL cases occur in northern countries, belonging to the Mediterranean basin and East Africa, where the main species are *L. infantum* and *L. donovani*, respectively [18].

In Algeria, a North African country, CVL is present throughout the country, with the prevalence of parasites varying depending on the region [19]. In the capital, Algiers, 37% of 1800 dogs had positive serology for CVL between 1990 and 1997 [20]. Also in Algiers, Adel et al. [21] showed in a study of 462 dogs conducted between 2004 and 2005 that stray dogs were those with the highest prevalence of leishmaniasis (11.7%), followed by animals from the national guard (9.7%), and those that lived on farms (5.9%), but there was no statistical difference among these groups. Between 2008 and 2009, there was an increase in the number of cases of CVL, from west to east, in the country’s coastal region, between the cities of Tlemcen and Jijel, which could be due to the increase in rainfall in the region, with some cities showing a statistical difference in the rise of CVL prevalence during this period [22]. Seropositive dogs were also found in Kabylia, a region located in the north, where the rate of seropositive animals was 36% [23], and in the Tiaret region, with 93/137 (67.88%) dogs testing positive for CVL through *Leishmania* nested PCR—*Ln*PCR [24].

The first reported case of canine infection in Morocco was published in 1932 [25]. CVL caused by *L. infantum* is present in the north and central–south regions [26]. Recently, 29 dogs infected with *L. infantum* have been found in northwestern Morocco, in urban centers such as Rabat (21/29 = 72.4%), Benslimane (2/29 = 6.9%), Casablanca (2/29 = 6.9%), Kenitra (1/29 = 3.45%), Fes (1/29 = 3.45%), Rommani (1/29 = 3.45%), and Jorf al Melha (1/29 = 3.45%) [27].

In Tunisia, the infection of dogs by *Leishmania* was first described by Nicolle and Comte [28]. As in other countries, the prevalence of CVL varies according to the region. In the city of Medjez El Bab, Beja Governorate, in northern Tunisia, the canine infection rate in 1994 and 1995 was 18% and 22.3%, respectively, and the positivity rate was based on serological tests, including both the enzyme-linked immunosorbent assay (ELISA) and the immunofluorescence antibody test (IFAT). It is noteworthy that 90% of the animals were asymptomatic [29]. In Sfax, the central–south region, 250 animals were tested for CVL, with a seroprevalence rate of 6% [30]. In the Kairouan Governorate, in the central region of the country, in 2005, the prevalence was 21% in dogs from the rural area of El Alaa, in which 50% of the animals were asymptomatic [31]. Similarly, in a study by Zoghlami et al. [32], in 2013, in seven districts of the same governorate, the overall prevalence was 26.7%. Currently, a seroprevalence of 58.3% was obtained in a study conducted by Bouattour et al. [33] in eight locations, in different bioclimatic areas of the country. According to the authors, it is possible to conclude that there was an increase in CVL in all bioclimatic areas, in addition to confirming the spread of the disease to arid areas.

In Egypt, asymptomatic infected dogs were found in five provinces in the north of the country, with a seroprevalence rate of 21.3%, with the highest being found in the provinces of Giza and Cairo [34].

In East Africa, the main species related to VL is *L. donovani*, and its transmission is considered mainly anthroponotic [35]. However, some studies have highlighted that dogs can be an important reservoir of this species in this part of the continent [35,36].

The existence of dogs infected with *Leishmania* in Ethiopia has been reported in several studies (Table 1). While in Sudanese territory, Dereure et al. [36] were the first to advance the possibility of the dog acting as a reservoir host in Barbar El Fugarra, in the state of Gedaref, eastern Sudan, which is considered an endemic area for VL. At the same location, a study conducted by Dereure et al. [37], involving dogs with poor body condition, observed a seroprevalence for leishmaniasis of 72.5% (37/51), 74.3% (26/35), and 42.9% (15/35) in the years 1998, 1999, and 2000, respectively. In contrast, a study by Hassan et al. [38], conducted in in 2002 in 10 villages along the Rahad River, located in eastern Sudan, showed that 6.9% of the dogs were serologically positive for *L. donovani.* In the western part of the continent, cases of dogs infected with *Leishmania*, mostly by *L. infantum*, have been reported in Burkina Faso [39], Côte d’Ivoire [40], Nigeria [41], and Zambia in southern Africa [42] (Table 1).

### 2.2. The Americas

In the Americas, the main etiological agent of CVL is *L. infantum*, which was introduced during the period of the Portuguese and Spanish colonization through infected animals, such as dogs, which came from Europe to the New World with the colonizers [43].

In Argentina, a country where the number of human VL cases rose in 2019, presenting nine cases (0.35% of cases in the Americans) [44], infected dogs were found in the city of Posadas, the province of Misiones [45,46], where Cruz et al. [47] reported a 57.3% infection rate of dogs infected by *L. infantum* (63/110). In this study, the dogs were diagnosed by serological tests (such as the rK39-immunochromatographic test (rK39-ICT) and IFAT), in addition to molecular testing (nested PCR). One group came from two shelters, while the other consisted of dogs owned by individuals who visited a local veterinary clinic, including both symptomatic and asymptomatic animals. In Puerto Iguazú, located in the same province, 7.17% of 209 dogs evaluated were considered positive for *L. infantum* [48], using rK39-ICT, IFAT, and nested PCR techniques. More recently, infected dogs were identified by Fujisawa et al. [49] and Lamattina et al. [50], in which 28 of 160 dogs (17.5%) were found with positive serology. Similarly, in the Chaco region of Salta Province, about 1000 km away from the province of Misiones, the prevalence of CVL was 13% [51].

In Uruguay, an autochthonous outbreak that occurred in Arenitas Blancas, in the department of Salto, 22% of the animals studied tested positive for leishmaniasis [52].

Reports dated from 1982 demonstrate the presence of CVL in Bolivia, where *L. infantum* was isolated from five dogs in the Yungas area, which was confirmed by serology [53].

In Colombia, VL has two sources of transmission: in the northern departments of Sucre, Bolívar, and Córdoba, where the highest number of cases are registered; and the Middle Valley region of the Magdalena River, between the central and eastern parts of the Colombian Andes mountain range, in the departments of Cundinamarca, Tolima, and Huila [54]. Picón et al. [55] found a 12% seroprevalence in 2016–2017, when analyzing dogs in the cities of Guamo, Ortega, Flandes, Coyaima, and Melgar in the Tolima department, and Neiva in the Huila department. Interestingly, Arbeláez et al. [56] reported the first case of urban CVL in the city of Cali, in the Valle del Cauca department, southwestern Colombia, in a dog that had never lived in an endemic area and only roamed in areas where human or canine cases of the disease had not been reported. More recently, dogs infected with *L. infantum* were found within the urban perimeter of Sincelejo. The frequency of dogs with CVL is as high as the frequencies found 8 years ago in the rural areas of this municipality, which is in the Department of Sucre [57], located in the Colombian Caribbean, as well as in Pradera and Florida, in Valle del Cauca, southwest Colombia [58].

In a study conducted in Venezuela, the seroprevalence of dogs infected with *L. infantum* was found to be 5.2% (3/58) in the state of Lara. In the state of Yaracuy, no seropositive dogs for *L. infantum*’s antigen were detected. However, qPCR testing identified one positive case [59]. In a study conducted with randomly sampled dogs from February to March 2019 on El Tigre Island, located in Amapala, Honduras, the seroprevalence of *L. infantum* infection was 41% (44/107), according to the dual path platform (DPP) rapid test and ELISA. Additionally, the presence of parasite DNA was detected in 94% of the dogs that tested positive in serological tests [60].

According to PAHO, 97% of all human VL cases that occurred on the continent in 2019 were reported in Brazil [7], and the disease continues to spread [61,62]. Cases of canine infection have been reported in several state capitals, or in their metropolitan regions, such as Belo Horizonte [63,64], Boa Vista [65], Cuiabá [66,67], Florianópolis [68], Niterói [69], Porto Alegre [70], and Vitória [71]. The same was observed in inland cities, such as Piacatu [72], in the state of São Paulo; Divinópolis [73] and Governador Valadares [74,75] in Minas Gerais; Campina Grande [76] in Paraíba; and Santarém [77], in the state of Pará. The seroprevalence of CVL in Brazil varies from 4% to 75%, being related to geographic conditions, climate, and social aspects of the affected area [78].

It is known that the disease has reached areas where it had previously not been present. The expansion of CVL to non-endemic regions can be illustrated by reports, such as the occurrence in the French overseas department of French Guiana. In 2016, dogs with autochthonous infection by *L. infantum* were reported in Cayenne through molecular tests, with a prevalence of 3.1%. In addition, it was observed in dogs that performed military work, which had negative molecular and serological results prior to going to French Guiana, yet tested positive for infection when they returned to France after four months in the region, suggesting the possible existence of an autochthonous cycle of VL transmission in the department [79].

### 2.3. Asia

VL on the Asian continent is caused by *L. infantum* in the Middle East and Central Asia and *L. donovani* in the Indian subcontinent, a region that harbors the most cases in the world [80,81,82].

In Iran, VL is distributed in three foci, located in four provinces in the northwest and south and, recently, it has spread to other parts of the country [83,84]. The prevalence of CVL in endemic areas ranges from 14.2% to 17.4% [83], and several studies have shown the distribution of infected dogs throughout Iran (Table 1).

After more than 30 years without the occurrence of human and canine infection, cases of CVL in Israel were reported by Baneth et al. [85]. Nasereddin et al. [86] isolated strains of Leishmania, mainly *L. infantum*, in dogs in the northern and central regions of the country, as well as in the West Bank, where 215 domestic dogs from seven districts were examined, and the overall prevalence of leishmaniasis was 16.7% [87]. Samples from 189 dogs, obtained in the town of Alfei Menashe, in the Shomron region, were analyzed and 3.3% of the animals tested positive for infection by *L. infantum* [88].

CVL in Turkey was first reported in 1951 in Istanbul and Bursa [89] and, according to Toz et al. [90], it is a common disease throughout the country, being mainly caused by *L. infantum*. Recently, in a study conducted by Koenhemsi et al. [91] in Istanbul, 5/171 (2.92%) dogs of different breeds and ages tested positive for CVL. Infected dogs in Central Asia have been found in Kazakhstan, Turkmenistan, and Uzbekistan [92]. Nepal [93] and Pakistan [94] have also identified dogs infected with the parasite (Table 1).

VL is a major public health issue in China, being considered endemic or re-emerging in six provinces, including Xinjiang, Gansu, Sichuan, Shaanxi, Shanxi, and Inner Mongolia. Zoonotic VL is endemic, with dogs acting as the main reservoir of *L. infantum* [95,96]. Shang et al. [95], using real-time PCR, were able to detect *Leishmania* DNA in dogs from Wenchuan (23/98 = 23.5%), Heishui (20/71 = 28.2%), and Jiuzhaigou (35/145 = 24.1%) counties, all located in Sichuan province, southwest China, where the total positivity rate was 24.8%. In 2022, Sandy et al. [97] were the first to report the occurrence of CVL in Hong Kong. Dogs infected with *Leishmania* have also been found in the Philippines and Vietnam [98].

### 2.4. Europe

For many years, leishmaniasis has been the only tropical vector-borne disease endemic in southern Europe, where the majority of reported cases are due to zoonotic VL caused by *L. infantum* [99].

Among the Mediterranean Basin countries, Albania is one of the most affected countries by VL, and since the late 1980s, several *L. infantum* isolates have been obtained from both human and canine cases [100]. In Bosnia and Herzegovina, domestic, stray, and shelter dogs from different parts of the country have been tested for leishmaniasis. In total, 16.7% of the animals tested positive, indicating that a significant part of the canine population was infected [101].

The county of Split-Dalmatia, in Croatia, is considered an endemic area for CVL, with seroprevalence ranging from 0 to 42.8%. Dogs from the city of Split and from 12 villages in the region were tested, and 14.7% of the animals in Split were positive for *L. infantum* infection, with eight villages having seroprevalence ranging from 7.1% to 42.8%, depending on their location [102]. In another study conducted in 2008, the seroprevalence was 31% in Split-Dalmatia and 3.8% in Šibenik-Knin county [103].

Cyprus, an island country located in the eastern Mediterranean region, did not register any cases of CVL for a period of more than 20 years due to a decline in the number of sandflies and dogs. However, there has been an increase in the population of vectors and dogs, due to the end of malaria and echinococcosis control programs, which directly affected the vector and dog population on the island, resulting in the resurgence of CVL in several areas in the south [104,105]. Infected dogs were also found in the island’s northern region, with rates ranging from 1.9% to 13.2% [106].

CVL is endemic in the south of France, where several foci have been identified, such as Alpes-Maritimes, Cévennes, and Provence [107,108]. A seroprevalence of 29.6% was reported in Cévennes, according to study by Lachaud et al. [109]. Dogs infected with *L. infantum* were found in several French cities, including Toulon and Hyères, in the department of Var, in addition to Miramas, Istres, and Salon-de-Provence, in the department of Bouches-du-Rhône, and Solenzara, on the island of Corsica [110]. Currently, through data obtained by veterinarians, it is possible to verify that the incidence of dogs infected by *L. infantum* in a large part of French territory is between 0 and 5 cases/1000 dogs/year. However, the number of cases in parts of south-eastern France has reached 20 cases/1000 dogs/year. Sporadic cases have been reported in the north of the country, which is considered a non-endemic region for the disease [111].

Between 2005 and 2010, samples from 5772 dogs from different parts of Greece were collected, with an average reported seroprevalence of 22.09%. It was shown to vary from 6.5% in western Macedonia to 50.2% on the island of Corfu, and of the 43 prefectures from which the canine samples were collected, 41 had seropositive dogs, showing that the disease is spreading [112]. According to studies conducted by Symeonidou et al. [113], it is possible to deduce that CVL remains a country-wide problem, since seropositive dogs were found throughout the Greek departments, and most prefectures had at least one seropositive animal, with seropositivity rates ranging from 0% to 53%. This variation in the prevalence of CVL cases in Greek territory is related to the varied geographical, climatic and socioeconomic conditions in the country, as well as the existence of dogs and vectors [113].

The main endemic areas for CVL in Italy include the central and southern regions and the islands of Sicily and Sardinia. In a study conducted on the island of Sardinia, involving dogs from a local kennel and those attending a veterinary hospital, the authors examined 1147 dogs, of which 15.4% (177/1147) were seropositive, similar to the seroprevalence of 17.7% previously reported for the mainland part of the country [114]. In recent decades, CVL has spread towards the north of the country, which had previously been considered a non-endemic area, with cases registered in Emilia-Romagna, Lombardia, Trentino-Alto Adige, Piedmont, and Vale D’Aosta [115,116,117,118,119]. Furthermore, the Republic of San Marino, an autonomous state located within Italian territory, presented a seroprevalence of 3.9% in 2012 [120], making the entire Italian Peninsula an endemic area for the infection [118].

In the Malta archipelago, 60 samples of dogs from different breeds were collected on the Islands of Malta and Gozo, and 20% tested positive for leishmaniasis through molecular testing [121].

In Portugal, according to Cardoso et al. [122], dogs testing positive for *L. infantum* are distributed throughout the mainland, namely the north, Central Region, Alentejo, Lisbon, and the Algarve. When analyzing 193 dogs living in kennels in the Algarve, Maia et al. [123], found a general CVL seroprevalence of 16.06%. Furthermore, between 2011 and 2014, 170 dogs were studied in the same region, and seroprevalence reached 18.2% [124]. Schallig et al. [125] showed that the number of seropositive dogs in Évora increased over a period of 20 years, with 9.4% in 1999 and 5.6% in 2010, as compared to 3.9% in 1990, the year of the initial study. In other study, conducted from May 2011 to February 2014, a total of 230 dogs from veterinary medical centers and animal shelters in southern Portugal were randomly separated in two groups. One analyzed group presented no clinical signs compatible with CVL, and the prevalence was 69% (107/155). In the other group, with clinical signs compatible with CVL, the prevalence was 42.7% (32/75) [126]. In contrast, between 2011 and 2012, 581 dogs from five shelters in Lisbon and Setúbal were analyzed, and 13.1% tested positive for *L. infantum* [127]. In a cross-sectional study by Almeida et al. [128], using samples from 1860 dogs from all parts of the country, the overall seroprevalence rate was 12.5%, with values ranging from 9.6% in the north to 17.2% in the Algarve.

CVL in Spain was considered restricted to the Mediterranean region. However, the disease has spread throughout most of the country, including the islands [129]. The CVL seroprevalence rate after a random analysis varies from 2% to 57.1% between regions [130], and these variations are related to environmental factors, geographic location, and vector dispersion [131]. Northern Spain is considered a non-endemic area, with low seroprevalence [130]. According to the results of a multicenter study by Diaz-Regañón et al. [132], the highest seroprevalence obtained in the north was found in the commune of Aragon (24.56%) and the lowest in Asturias (1.27%). Through a cross-sectional serological survey conducted between 2011 and 2016, Galvéz et al. [130], showed a high rate of seroprevalence in the southern Spanish provinces, such as Malaga (29.4%), Seville (25%), Murcia (23.7%), and the Balearic Islands (20%). In addition, for the first time, the seroprevalence of *L. infantum* on the Canary Islands was reported, with 2.45% of dogs also testing positive for leishmaniasis [129].

Cases of CVL have been reported in other European countries, such as Bulgaria [133], Georgia [134,135], Germany [136], Hungary [137], Kosovo [138], the Netherlands [136], North Macedonia [139], Romania [140,141,142], Slovenia [143], Switzerland [136], and the United Kingdom [144] (Table 1).

**Table 1 pathogens-13-00455-t001:** Percentage of CVL positive dogs and *Leishmania* species described in African, Asian, American, and European countries.

Continent	Country	Location	Parasite Species	% of Dogs Positive for *Leishmania* (Diagnostic Test Used)	References
Africa	Ethiopia	Amhara region	*L*. *donovani* Complex *	3.8% (RIFI and ELISA); 2.8% (PCR)	[145]
		Kafta Humera district	*L. donovani*	27.7% (DAT); 14.8% (KDRT)	[146]
		Addis Zemen, Humera, and Sheraro	*L. donovani* Complex *	5.9% (PCR)	[147]
		Benishangul-Gumuz region	NA	13.9% (rK39 ICT); 5.6 (DAT)	[148]
	Burkina-Faso	Bobo-Dioulasso	*L. infantum*	5.88% (Serological test)	[39]
	Côte d’Ivoire	Abidjan and Yamoussoukro cities	*L. infantum*	8.9% (PCR or IFAT)	[40]
	Nigeria	Oyo, Ogun, and Kwara	NA	4.4% (ELISA)	[41]
	Zambia	Southern Province	*L. infantum*	NA (Serological tests and PCR)	[42]
Asia	Iran	Meshkin-Shahr district	*L. infantum*	15.8% (DAT)	[149]
		Boyer Ahmad district	*L. infantum*	10% (DAT)	[150]
		Kouhsar district	*L. infantum*	3.6% (DAT)	[151]
		Khorasan Razavi province	*L. infantum*	7.6% (IFAT)	[152]
		Kerman province	*L. infantum*	11.25% (DAT)	[153]
		Kerman and Sistan-Baluchestan provinces	*L. infantum*	15.4% (ELISA)	[154]
		Tehran and Alborz provinces	*L. infantum*	4.9% (DAT)	[155]
		Meshkin-Shahr district	*L. infantum*	23.4% (DAT)	[156]
		Hamedan province	*L. infantum*	3.95% (ELISA)	[157]
		Jiroft district	*L. infantum*	7.9% (DAT)	[158]
		Ardabil, Alborz, and East-Azerbaijan provinces	*L. infantum*	100% (DAT, rK39, and PCR)	[84]
		Alborz province	*L. infantum*	2.97% (DAT)	[159]
		Golestan province	*L. infantum*	18% (PCR	[160]
	Nepal	Kathmandu	*L. donovani* Complex *	18.57% (PCR)	[93]
	Pakistan	Chilas, Abbotabad, Bagh, Poonch, and Muzafarabad districts	*L. donovani* Complex *	18% (DAT); 26.6% (ELISA)	[94]
	Philippines	NA	*L. infantum*	NA	[98]
	Vietnam	NA	*L. infantum*	NA	[98]
America	Brazil	Belo Horizonte	*L. infantum*	64.6% (IFAT)	[63]
			*L. infantum*	56.7% (PCR)	[64]
		Boa Vista	*L. infantum*	10.3% (RIFI)	[65]
		Baixada Cuiabana	*L. infantum*	16.58% (PCR)	[66]
		Cuiabá	*L. infantum*	1.14% (DPP and ELISA)	[67]
		Florianópolis	*L. infantum*	1.37% (ELISA and RIFI)	[68]
		Niterói	NA	15.5% (ELISA and IFAT)	[69]
		Porto Alegre metropolitan area	NA	4% (PCR)	[70]
		Vitória	NA	13% (ELISA); 6% (RIFI)	[71]
		Piacatu	*L. infantum*	16.08% (ELISA)	[72]
		Divinópolis	NA	4.63% (ELISA and IFAT)	[73]
		Governador Valadares	NA	30.2% (IFAT)	[74]
			NA	29% (ELISA and RIFI)	[75]
		Campina Grande	NA	8.4% (IFAT) and 4.3% (ELISA)	[76]
		Santarém	*L. infantum*	23.3% (ELISA and rK39)	[77]
Europe	Bulgaria	Petrich	*L. infantum*	NA (IFAT and PCR)	[133]
	Georgia	Tbilisi and Kutaisi	*L. donovani* Complex *	20% (rK39)	[134]
		Kvareli and Sagarejo districts		19.5% and 11.4% (rK39)	[135]
	Germany	NA	NA	11.8% (ELISA)	[136]
	Hungary	Tolna province	*L. infantum*	30% (IFAT and PCR)	[137]
	Kosovo	Prizren, Gjakova, Rahovec, and Deçan	*L. infantum*	18.49% (ELISA)	[138]
	The Netherlands	NA	NA	32.4% (ELISA)	[136]
	North Macedonia	Skopje and Prilep	*L. infantum*	2.5% (PCR)	[139]
	Romania	Vâlcea County	*L. infantum*	NA (FASTest^®^LEISH)	[140]
		Ramnicu Vâlcea	*L. infantum*	8.75% (ELISA); 10% (PCR)	[161]
		Galați	NA	8.33% (ELISA)	[141]
		Argeș County	*L. infantum*	20.1% (PCR)	[142]
	Slovenia	Kostelo	*L. infantum*	NA (IFAT)	[143]
	Switzerland	NA	NA	12.2% (ELISA)	[136]
	United Kingdom	NA	*L. infantum*	NA (ELISA and PCR)	[144]

IFAT: Indirect immunofluorescence reaction; ELISA: enzyme-linked immunosorbent assay; DAT: direct agglutination test; PCR: Polymerase Chain Reaction. NA: not applicable. * Classification that encompasses the species *L. donovani*, *L. infantum* (Old World), and *L. chagasi* (New World). Unfortunately, the majority of studies do not provide the sensitivity and specificity of the tests used for CVL diagnosis/detection. Exceptions include researches 66 [PCR—sensitivity (S): 94.7%, specificity (E): 100% (bone marrow); S: 91.8%, E: 100% (lymph node); S: 98%, E:100% (blood)] and 141 [ELISA—S: 95%, E: 96%].

## 3. Factors Influencing the Territorial Expansion and Transmission Maintenance of Visceral Leishmaniasis

It is estimated that about 2.5 million dogs are infected on the European continent, also reaching millions in South America [162]. Furthermore, this number is believed to have increased mostly in the northern and eastern hemispheres [161]. Many factors contribute to the spreading of the disease, especially those related to environmental and demographic changes, such as long periods of drought, global warming, deforestation, urbanization, and migration [5,161,163,164] (Figure 1).

Environmental changes, such as global warming and deforestation, can affect the distribution of leishmaniasis in several ways. Among them are changes in vector biology, such as the effect of temperature on parasite development, vector competence [165,166], an increase in the vectors’ breeding season in a given area, and their presence in areas where they were once absent [166,167,168,169,170]. According to Santos et al. [171], the probabilities of the occurrence of the vector and cases of CVL and HVL in deforested areas were, respectively, 2.63, 2.07, and 3.18 times greater, as compared to non-deforested areas. Socioeconomic impacts are also an aggravating factor in spreading the disease. Over the years, several countries have suffered from a rise in the prevalence of neglected tropical diseases due to human migration from countries suffering from war and famine [172], as well as migration provoked by climatic factors, such as the one that occurred in Northeastern Brazil due to a prolonged drought, compelling many people to leave the countryside and settle in large cities, mainly on the outskirts [163]. Problems in accessing basic care in areas of conflict, such as during a civil war, as reported in Somalia and South Sudan [173,174], also present formidable challenges.

Living on the outskirts of an urban center offers a favorable environment for the occurrence of new outbreaks of CVL transmission, as it is often related to areas of recent habitation, close to open fields and wooded areas. Therefore, the disorganized occupation of this space, precarious living conditions, and the lack of basic infrastructure, in addition to frequent contact with domestic animals, especially infected dogs, create an ideal environment for the VL transmission cycle [164,175,176], which is often related to the houses that have gardens and backyards, where dogs are generally kept [177,178].

The presence of trees, birds, shade, and animal feces in these areas attracts sandflies, facilitating their contact with dogs [72,78,179]. According to the findings by Martín-Sánchez et al. [180] and Coura-Vital et al. [181], dogs that sleep outdoors are more likely to be infected, as compared to those that spend most of their time indoors. Studies have also shown the importance of stray dogs in the epidemiology of VL [81,151,153,158,179,182]. According to Babuadze et al. [134], the errant behavior of these dogs is a contributing factor to the spread of leishmaniasis into new areas.

It has been suggested that CVL normally precedes the appearance of cases in humans [183,184,185] and, according to some authors, there are no reports of infection in humans without the presence of infected dogs [73,175]. Therefore, the spatial overlap of human and canine VL cases in urban areas is an important condition in transmitting the disease to humans [75,119,186,187,188].

Some characteristics of dogs appear to favor *Leishmania* infection, such as breed, age, coat, and size. Studies have shown that Boxer, German Shepherd, Doberman, Foxhound, and Beagle breeds are among those most affected by *Leishmania* infection and disease progression [189,190,191,192,193]. Older animals had higher seroprevalence in some studies, and this may be related to the time of exposure to the vectors, as the older the animal is, the more contact with the vector it has [95,156,157,158]. Regarding the coat, the blood meal seems to be easier in dogs with short coats, which give the sandfly better access to the animal’s skin. In addition, it may be related to higher CO_2_ production, attracting vector insects [70,78,190,194]. This is also observed in larger dogs, as they have a larger contact area, making them more susceptible to bites [38,130,195]. In relation to sex, several authors found no difference between males and females regarding the risk of infection [152,153,159,181,196,197,198].

Thus, assorted factors are associated with the occurrence and spread of CVL, especially in urban and peri-urban areas. However, they must be more thoroughly studied to better understand the epidemiology of the disease.

## 4. The Dog’s Role in the Visceral Leishmaniasis Transmission Cycle

Dogs with visceral leishmaniasis (VL) play a key role in the transmission of protozoan parasites of the *Leishmania donovani* complex in urban areas. Deane [199] and Deane and Deane [200] contributed to the understanding of the epidemiology of *L. infantum*, documenting the participation of the dog as a domestic reservoir and the fox as a wild reservoir of the parasite. In this sense, there is a substantial overlap between locations where human cases have been reported and high canine seroprevalence, highlighting the close relationship between canine and human infections [184,201]. Most infected animals are asymptomatic (clinically healthy), but they can host the parasite in the skin [13,202]. In symptomatic dogs, the phlebotomine sandflies’ capacity for infection is greater when compared to asymptomatic dogs due to the high parasite load on the animals’ skin, facilitating infection. When evaluating the disease transmission potential in relation to the parasite load, Courtenay et al. [203] showed that, in sick and infectious animals, the spread of parasites to the skin was greater, especially in the ear skin, suggesting that this is the most infectious region for the sandflies. Thus, the dog represents an abundant source of the parasite to invertebrate hosts, being found in all foci of human disease, and is characterized as the main link in the VL transmission chain [204].

Michalsky et al. [205] showed that symptomatic animals had an average rate of sandfly infectivity of 83.3%, with 28.4% of the vectors being infected. In contrast, the rates of oligosymptomatic and asymptomatic animals were 16.7% and 33.3%, with the percentages of infected vectors at 5.1% and 5.4%, respectively. Verçosa et al. [206] also obtained similar results, proving the high parasite load in ear tissues. In this test, using cytological impressions from 22 different sites of the animal, it was observed that symptomatic dogs had amastigote forms of the parasite in at least one analyzed region. Conversely, no amastigote form was found in asymptomatic individuals. In addition, the study found that six-ninths of the symptomatic dogs were able to transmit the parasite to insect vectors. However, in contrast to the results obtained by Michalsky et al. [207], no asymptomatic animal was able to infect sandflies.

Although some studies consider that symptomatic animals are the main reservoirs, the importance of asymptomatic animals in the transmission of the parasite has been supported by studies such as those by Molina et al. [208], who even claimed that the clinical status of dogs did not influence the infectivity of *P. perniciosus*—three out of five asymptomatic dogs were able to transmit *L. infantum* to sandflies. In fact, the clinical status of the animals did not interfere with the infection of the vectors, according to Guarga et al. [207]. However, the percentage of females that fed slightly decreased as the clinical signs in the dogs increased, suggesting a possible predilection for healthy skin. Other studies, such as those by Solano-Gallego et al. [209], using bone marrow, conjunctiva, skin PCR, and ELISA, proved that asymptomatic animals could also be considered reservoirs of *L. infantum*. Corroborating these findings, Moshfe et al. [210] observed that 43.4% of the dogs without clinical signs had *L. infantum* DNA in peripheral blood, detected by PCR, while Coura-Vital et al. [211] showed for the first time that asymptomatic dogs were seronegative, yet PCR positive. According to the authors, these animals presented twice the risk of seroconversion, as compared to dogs with negative PCR results. More recently, in an endemic region of the state of Minas Gerais, Brazil, the parasite in 73% of asymptomatic dogs was found mainly on the skin [212].

Therefore, dogs carrying the parasite should be considered infectious for the vectors, regardless of any clinical signs [158,208]. In addition, these animals play an important role in the epidemiology of VL, since a large number of animals with skin parasitism is able to infect sandflies and thus perpetuate disease transmission in urban and peri-urban areas [188,213].

## 5. The Euthanasia Challenges for Controlling the Spread of Canine Visceral Leishmaniasis

In many endemic countries, the elimination of infected dogs has been recommended as a means of controlling VL [214]. In South America, euthanasia is officially recommended in Brazil, Colombia, and Venezuela. In addition, it is also recommended in Morocco and Tunisia, the Mediterranean region, and some Middle Eastern countries, such as Iran, Iraq, and Syria, as well as in Armenia, Azerbaijan, and Uzbekistan, in Central Asia [2,215]. Furthermore, in some countries, this practice is not officially recommended, although it is nonetheless carried out, for example, in Afghanistan, Albania, Saudi Arabia, Algeria, Kazakhstan, China, Georgia, Paraguay, and Tajikistan. On the European continent, euthanasia is generally performed on animals with a serious form of the disease, as a way to prevent prolonged suffering [2,215].

The practice of euthanasia is controversial among researchers, who still discuss its effectiveness in eliminating VL, not to mention it being strongly criticized from an ethical perspective, with resistance from dog owners and the professionals responsible for carrying out this work [216,217].

Furthermore, this practice faces other challenges. For instance, the limitation of serological tests should be highlighted, as shown by Grimaldi et al. [218], when evaluating the sensitivity of the rapid DPP test, which showed low sensitivity in asymptomatic dogs, with a rate of 47%, reaching 98% among the symptomatic animals. Serological tests, such as IFAT and ELISA, are also widespread but have cross-reaction problems and can generate false-positive and false-negative results [219]. This could lead to the improper elimination of dogs, while allowing infected animals to remain in the transmission cycle of the parasite [215,219].

The delay between diagnosis and the removal of animals is another issue and, according to Dantas-Torres et al. [215], newly infected dogs may remain undetected for months before a new investigation is conducted. Seronegative animals may be infected but have not yet undergone seroconversion, with these dogs roaming around freely and acting as a source of parasites for vector insects [220]. In fact, seronegative dogs displaying PCR positivity in the skin have previously been reported, but their epidemiological role remains unclear [184,211], and their potential infectivity is theoretically possible.

The replacement of euthanized dogs with other susceptible dogs also influences the effectiveness of euthanasia [215,220,221], as shown in studies by Moreira et al. [222] and by Nunes et al. [216] in endemic areas of Brazil.

Nevertheless, studies such as those by Ashford et al. [223], Costa et al. [224], Nunes et al. [225], Costa et al. [226], and Bermudi et al. [227], have shown that the elimination of infected dogs reduces the incidence of VL in humans and dogs. Conversely, some researchers, such as Dantas-Torres et al. [215], Dietze et al. [228], Paranhos-Silva et al. [229], and Vaz et al. [230], have shown the opposite.

In Brazil, thousands of dogs are eliminated every year and, even so, the prevalence of CVL remains high in several endemic foci. In fact, animal euthanasia is not capable of controlling zoonotic VL in the country [215]. This scenario corroborates a study by Costa et al. [231], which demonstrated that the elimination of dogs alone is not effective in areas of high transmission. However, dog screening and faster culling intervention (30 days after CVL diagnosis) during two years of VL transmission in the north of Minas Gerais State (Brazil) reduced the number of human cases by 75% and canine disease, cases as recently reported [232].

## 6. Discussion

The dog is considered a key element in the VL cycle caused by *L. infantum*, especially in the Americas and the Mediterranean basin.

The paucity of studies and surveys on the prevalence of CVL is noteworthy, as many regions have no data on it. As shown, CVL has been spreading into new territories due to various factors. One example is the migratory flow of dogs, which often travel to endemic areas with their human guardians or are bought in these areas, becoming a risk for the emergence and transmission of the disease in places that had not been considered endemic for VL [168,233].

According to Figueiredo et al. [234], some precautions can be taken to minimize the expansion, such as stricter control of the flow of these animals in endemic areas, mandatory serological screening of dogs that are traveling, and mandatory notification of positive cases. However, few countries have mandatory reporting for CVL [235].

Another factor to consider is the effectiveness of the detection methods. Could it be that the dogs in the cited studies were not positive or was it a reflection of the diagnostic method? Studies by Carvalho et al. [236] and Pessoa-e-Silva et al. [237] compared several serological and molecular methods used in the diagnosis of CVL and showed that the percentage of agreement between the tests varies from low to moderate. The efficiency of serological tests depends on several factors, especially the antigen used [218]. According to Travi et al. [214], the rK39 antigen was the most successful in the transition from the academic to the clinical environment, as it demonstrated high sensitivity and was able to detect an active infection. However, it does not have good specificity and sensitivity to identify asymptomatic infections, as compared to other serological methods (Table 2). Although molecular tests also present problems, they are better at detecting the parasite compared to serological tests [184,238]. Therefore, the number of infected dogs in some areas could be considered underestimated, since the accuracy of diagnostic methods is limited.

Controlling the canine population is also extremely important in the epidemiology of CVL. Countries such as Argentina [245], Brazil [246], Colombia [247], Paraguay [248], Uruguay [249], among others in South America, recommend the euthanasia of *L. infantum*-infected dogs. However, as already shown, the practice of euthanasia is controversial for ethical and social reasons, in addition to being considered costly [250].

Thus, in addition to the control measures recommended by the Brazilian Ministry of Health, a vaccine for prophylactic purposes was commercially available until 2015 [251]. The sale of LeishTec^®^, the only vaccine available in Brazil, was suspended in 2023. The effectiveness of vaccines currently on the European market varies. CaniLeish^®^ prevents clinical signs in 68.4% of cases and provides a protection level—defined as the percentage of vaccinated animals that remain asymptomatic (and not in terms of parasite protection)—of 92.7%. Similarly, LetiFend^®^ has shown a 72% effectiveness in preventing clinical signs and helps reduce these signs in infected animals [252,253]. The levels of protection provided by these vaccines are not sufficient to prevent infection by *L. infantum* [17,251,254]. In fact, as shown in a study carried out by Cotrina et al. [253], dogs vaccinated with LetiFend^®^ showed a 9.4% positivity rate for *L. infantum*, with no statistical difference compared to the control group (16.1%). For CaniLeish^®^, Oliva et al. [252], through a randomized double-blind controlled trial conducted in an endemic area for CVL, showed that vaccinated dogs presented 50.22% positivity for *Leishmania*, while the control group presented 66.66%. Furthermore, even when vaccinated, dogs can be a source of parasites for sandflies and, consequently, for other dogs and humans [255]. The treatment also has drawbacks, since the parasitological cure of infected animals is not achieved, and there is a high rate of recurrence and the possibility of resistance to the drugs [17,256]. Nonetheless, these alternatives have been used together with repellent collars, which have been widely accepted by dog owners in Europe [168] and are highly regarded for their effectiveness, according to studies by Gavgani et al. (Iran) [187], Sevá et al. (Brazil) [217], and Ribas et al. (Brazil) [257]. In a mathematical model, Sevá et al. [217] found that applying collars to 90% of dogs effectively eliminated the disease among humans and nearly wiped out the infection in the dog population as well. In 2021, the Brazilian Ministry of Health made the use of collars official as a way of controlling the disease in municipalities where transmission varies from high to very intense [258]. Furthermore, the sandflies antigens have been considered a promising strategy in blocking CVL transmission [17,251,254,256]. As such, there is a pressing need to find new methods and protocols to reduce limitations in the prevention, treatment, and control of CVL. Improvements in technology that facilitate the development of vaccines capable of disrupting the transmission of *L. infantum* from infected dogs could serve as a highly effective measure in preventing its spread. In fact, some studies have described how sandfly antigens interfere with the vector life cycle and help control de *L. infantum* infection after a blood meal. This is compatible with de hypothesis that effective control of CVL transmission can be achieved [17,251,254,256,259,260].

## 7. Conclusions

Based on the information presented in this review, it becomes evident that dogs play a pivotal role in the transmission cycle of visceral leishmaniasis caused by *L. infantum*. However, it is crucial to acknowledge that various other factors contribute significantly to this transmission dynamic. Environmental conditions, socioeconomic factors, climatic variations, as well as shortcomings in disease treatment and control efforts, all play significant roles in shaping the prevalence and spread of the disease. More comprehensive studies are needed to better understand these factors and find ways to overcome these failures.

## Figures and Tables

**Figure 1 pathogens-13-00455-f001:**
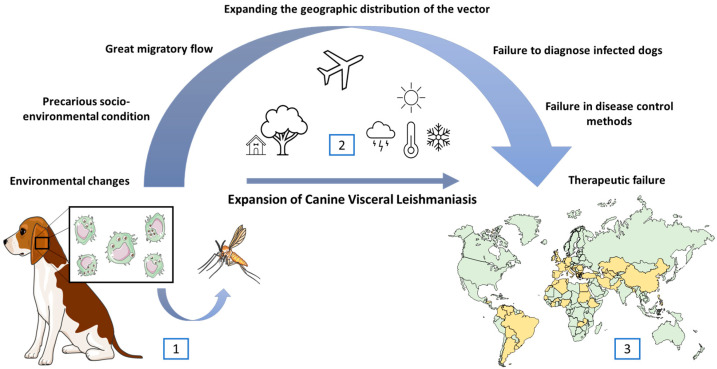
The dog is considered the main reservoir of *Leishmania infantum*, as the presence of amastigotes of the parasite in its skin favors the infection of sandfly vectors and, consequently, the maintenance of the disease (1). Several environmental (global warming, deforestation), social (migration), and economic factors (poverty, lack of basic sanitation) are related to the permanence of CVL in endemic areas (2), as well as its expansion into regions where it had not previously existed. The countries represented in yellow on the map are those that have data in this review (3).

**Table 2 pathogens-13-00455-t002:** Sensitivity and specificity of serological tests used for CVL diagnosis.

Test	Manufacturer	Sensitivity	Specificity	References
IFAT 1:40	Bio-Manguinhos, Rio de Janeiro, Brazil	96%	18% *; 76% **	[239]
IFAT 1:80		90%	33% *; 93% **	
ELISA	Bio-Manguinhos, Rio de Janeiro, Brazil	91%	79% *; 98% **	[239]
		84.20%	95.6%	[240]
rK39 (Kalazar Detect)	InBios, Inc., Seattle, WA, USA	88%	74% *; 98% **	[239]
		82.90%	92.6%	[241]
		76.90%	98.6%	[242]
		79.60%	95.7%	[240]
FAST	Royal Tropical Institute, Amsterdam, The Netherlands	93%	68% *; 100% **	[239]
DAT	Royal Tropical Institute, Amsterdam, The Netherlands	96%	33% *; 98% **	[239]
TR DPP	Bio-Manguinhos, Rio de Janeiro, Brazil	98%	60% *; 98% **	[239]
		21.74%	92.59%	[243]
		89%	70.2%	[244]
		85.70%	79.6%	[241]
		93.70%	95.9%	[242]
		97.90%	93.6%	[240]

IFAT: indirect immunofluorescence reaction; ELISA: enzyme-linked immunosorbent assay; DAT: direct agglutination test; FAST: fast agglutination screening test; TR DPP: dual path platform fast test. * Value referring to high transmission area and **, value referring to low transmission area, according to the authors.

## Data Availability

All data collected were reported in the text.

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
