# Peer review of "Global Distribution of Canine Visceral Leishmaniasis and the Role of the Dog in the Epidemiology of the Disease"

_pathogens, 2024, doi:10.3390/pathogens13060455_

Round 1

Reviewer 1 Report

Comments and Suggestions for Authors

The submitted paper is well driven and composed. However, there are some incongruences that need clarification or correction:

-Line 55: current data indicate that India is not anymore one of these 10 top countries. Please correct it.

-Line 56: of those countries reporting VL cases. Please include it.

-Line 59: caused mainly by. Please include it.

-Line 65: is this rate standard or an outlier?, indicate a range or more studies. It seems very high.

-It should be introduced the different techniques used to measure prevalence, and their specificity and sensibility (range) to understand results from table 1.

-Line 78. It is also endemic for cutaneous leishmaniasis. Please, reformulate the sentence.

-Line 85. Is this difference statistically significant?. Please include it.

-Line 87: please cite the factor involved: outbreak, climate,..Was it statistically significant?. Please include it.

-Line 99: Is Medjez El Babhow in the north?, how was measured this infectivity? Please include it.

-Line 103. Indicate the date of both studies for comparison.

-Line 118: where Barbar El Fugarra  is located?, North of the country?, Please include it.

-Line 119: was it a study for estimating the seroprevalence in random dogs or with clinical signs of leishmaniasis?, the rates are very high. This question should be taken into account when describe all the result of the review.

-Line 120: where the villages along the Rahad River are located?, West of the country?. Please include it.

-Results from study 175 seems to be affected by potential inespecificity of the technique. Similarly, results from study 42 could potentially be inaccurate (never have I see this figure before). It is also extrange the results from study 178. Is there any detail regarding these studies that could be introduced to clarify these outlier results?. Are these studies well designed?

-Please include in Table 1 the studies from The Americas. In addition, other studies are missing in that Table. Please complete Table 1 or include all the studies cited in a supplementary table.

-Line 131: What technique was used for studies 48 and 49?, Please include it.

-Line 131: was it (48) a study for estimating the seroprevalence in random dogs or with clinical signs of leishmaniasis?, the rates are very high. As previously asked, this question should be taken into account when describe all the result of the review.

-Line 141: indicate incidence (highest number of cases). Maybe you should describe difference of incidence and prevalence at some point, introduction?

-Line 142 and 144: where Middle Valley region of the Magdalena River is located?, Neiva in the Huila departments?, Please include it.

-Line 146: where Cali is located?, Please include it. It is very difficult to follow the rationale of the prevalence without a map or geographical details.

-Line 148: were no description of dogs infected with L. infantum before?. where is located?, Please include it. It is very difficult to follow the rationale of the paper without a map or geographical details.

-Line 159: was it a study for estimating the seroprevalence in random dogs or with clinical signs of leishmaniasis?, the rates are very high. This question should be taken into account when describe all the result of the review.

-Line 174: indicate prevalence and if this was autochthonous infection or imported infection.

-Line 189: where is located?, Please include it.

-Line 196 and 199: data showed the prevalence and incidence in the past. Current data indicate that India is not anymore in those figures after the kala-azar elimination program. Please correct it.

-Line 202. On Oct 31, 2023, WHO announced that Bangladesh has eliminated visceral leishmaniasis (also known as kala-azar) as a public health problem, the first country worldwide to do so. This elimination status was validated by WHO after Bangladesh reported less than 1 cases per 10 000 people in the previous 3 consecutive years. Please correct it.

-Line 203: What technique was used for studies 99, 115 and 116?, Please include it. Why are not included in Table 1?

-Line 210: indicate the prevalence in China and technique used to measure prevalence.

-Line 233: please cite the factor involved: climate,..

-Line 253: references are lacking, please include them. Cite the factor involved: climate, ..

-Line 254: indicate a range of prevalence and if it is affected by geographical distribution.

-Line 262: why is it important?, geographical area?. What about Italian Alps?

-Line 275: was it a study for estimating the seroprevalence in random dogs or with clinical signs of leishmaniasis?, the rates are very high. This question should be taken into account when describe all the result of the review.

-Line 283: was it a study for estimating the seroprevalence in random dogs or with clinical signs of leishmaniasis?, the rates are very high. This question should be taken into account when describe all the result of the review.

-Line 403: please confirm that 43.4% of dogs, without clinical sign, were positive for PCR. It is a high rate in blood for asymptomatic dogs. If so, was it influenced by some factors?

-Line 458: how much? Significantly?

-In my opinion, paragraph 5 only described elimination of infected dogs. I miss information about vaccination and collars, important tools for controlling leishmaniasis. Introduce please some studies about vaccination and collars, or change the title of the paragraph.

-Line 462: humans are also a source of infection for sandflies, please correct.

-Line 476: It seems reasonable to comment here some of the outliers described previously, example study 178 among others.

-Line 493: Brazil is the only country cited. Other general measures from other countries should be included.

Does any country have a national plan including dog control?.

-Line 501: indicate levels of induced protection and differentiate them from capability to infect.

-Line 508: indicate levels of effectiveness.

-Line 509: from where?, like in line 493 only Brazil?

-Line 514: new technologies?, give examples.

-Line 518: Is including here anthroponotic countries?, Although there might be transmission caused by the dogs in these countries, do you conclude that the dog is the main actor there?

-Line 521: is canine visceral leishmaniasis being spread due to poverty?

Comments on the Quality of English Language

None

Author Response

Reviewer #1:

Comments and Suggestions for Authors

The submitted paper is well driven and composed. However, there are some incongruences that need clarification or correction:

-Line 55: current data indicate that India is not anymore one of these 10 top countries. Please correct it.

We acknowledge the Reviewer #1 for the critical reading of the manuscript. We agree with the suggestion, and the revised sentence was clarified as follows:

“…It is estimated that 50,000 to 90,000 new cases of VL occur annually worldwide, with the majority occurring in Brazil and East African countries [5]. …”

-Line 56: of those countries reporting VL cases. Please include it.

We accepted the suggestion and the sentence appears in revised manuscript as follows:

“… In the Americas, LV is present in more than 10 countries, such as Argentina, Bolivia, Brazil, Colombia, El Salvador, Guatemala, Honduras, Mexico, Paraguay, Uruguay and Venezuela…”

-Line 59: caused mainly by. Please include it.

We accepted the suggestion and the sentence appears as:

“… Leishmania (Leishmania) donovani is the species responsible for VL in Africa and Asia, while in the Mediterranean region and the Americas, the disease is caused mainly by Leishmania (Leishmania) infantum (syn = Leishmania chagasi) [6,8,9]…”

-Line 65: is this rate standard or an outlier?, indicate a range or more studies. It seems very high. That rate is an outlier, a place with high transmission.

We are thankful for this comment. We have made appropriate changes in the sentence that appears in revised version of the manuscript as follows:

…In some regions, especially in South America the seroprevalence can reach 33% in Margarita Island, Venezuela and 36% in Jacobina, Brazil [13]…”

Moreover, the original reference 13 was replaced to: Moreno, J.: Alvar, J.; Canine leishmaniasis: epidemiological risk and the experimental model. Trends Parasitol 2002, 18, 399-405, doi:10.1016/s1471-4922(02)02347-4.

-It should be introduced the different techniques used to measure prevalence, and their specificity and sensibility (range) to understand results from table 1. 

Thanks for your suggestion. The methods used to measure prevalence are presented in Table 1 in the column entitled “% of dogs positive for Leishmania (diagnostic test used)”, when not reported in the article it is represented by “NA: Not applicable”. The specificity and sensitivity data of all the cited papers in the Table 1 were checked. Unfortunately, it is not possible to add the performance of the tests in most cases, given the studies used commercial tests in the referenced articles without the description of specificity and sensitivity data.

We have made appropriate changes in the Table 1 including the data missing in the Table 1 that appears in the revised manuscript as follows:

Table 1. Percentage of CVL positive dogs and Leishmania species described in African, Asian, American and European countries.

Continent

Country

Location

Parasite species

% of dogs positive for Leishmania (diagnostic test used)

References

Africa

Ethiopia

Amhara region

L. donovani Complex*

3.8% (RIFI – S: NA, E: NA ; and ELISA– S: NA, E: NA); 2.8% (PCR– S: NA, E: NA)

[148]

Kafta Humera district

L. donovani

27.7% (DAT– S: NA, E: NA); 14.8% (KDRT– S: NA, E: NA)

[149]

Addis Zemen, Humera, and Sheraro

L. donovani

Complex*

5.9% (PCR– S: NA , E: NA)

[150]

Benishangul-Gumuz region

NA

13.9% (rK39 ICT– S: NA, E: NA); 5.6 (DAT– S: NA, E: NA)

[151]

Burkina-Faso

Bobo-Dioulasso

L. infantum

5.88% (Serological test– S: NA, E: NA)

[39]

Côte d'Ivoire

Abidjan and Yamoussoukro cities

L. infantum

8.9% (PCR – S: , E: or IFAT– S: NA, E: NA)

[40]

Nigeria

Oyo, Ogun, and Kwara

NA

4.4% (ELISA– S: NA, E: NA)

[41]

Zambia

Southern Province

L. infantum

NA (Serological tests – S: NA, E: NA and PCR)

[42]

Asia

Iran

Meshkin-Shahr district

L. infantum

15.8% (DAT– S: NA, E: NA)

[152]

Boyer Ahmad district

L. infantum

10% (DAT– S: NA, E: NA)

[153]

Kouhsar district

L. infantum

3.6% (DAT– S: NA, E: NA)

[154]

Khorasan Razavi province

L. infantum

7.6% (IFAT– S: NA, E: NA)

[155]

Kerman province

L. infantum

11.25% (DAT– S: NA, E: NA)

[156]

Kerman and Sistan-Baluchestan provinces

L. infantum

15.4% (ELISA– S: NA, E: NA)

[157]

Tehran and Alborz provinces

L. infantum

4.9% (DAT– S: NA, E: NA)

[158]

Meshkin-Shahr district

L. infantum

23.4% (DAT– S: NA, E: NA)

[159]

Hamedan province

L. infantum

3.95% (ELISA– S: NA, E: NA)

[160]

Jiroft district

L. infantum

7.9% (DAT– S: NA, E: NA)

[161]

Ardabil, Alborz, and East-Azerbaijan provinces

L. infantum

100% (DAT– S: NA, E: NA, rK39– S: NA, E: NA, and PCR– S: NA, E: NA)

[87]

Alborz province

L. infantum

2.97% (DAT– S: , E:)

[162]

Golestan province

L. infantum

18% (PCR– S: , E:)

[163]

Nepal

Kathmandu

L. donovani

Complex*

18.57% (PCR– S: NA, E: NA)

[96]

Pakistan

Chilas, Abbotabad, Bagh, Poonch, and Muzafarabad districts

L. donovani

Complex*

18% (DAT– S: NA , E: NA); 26.6% (ELISA– S: NA, E: NA)

[97]

Philippines

NA

L. infantum

NA

[101]

Vietnam

NA

L. infantum

NA

[101]

America

Brazil

Belo Horizonte

L. infantum

64,6% (IFAT– S: NA, E: NA)

[63]

L. infantum

56,7 % (PCR– S: NA, E: NA)

[64]

Boa Vista

L. infantum

10,3% (RIFI– S: NA, E: NA)

[65]

Baixada Cuiabana

L. infantum

16,58% (PCR– S: 94,7%, E:100% (bone marrow); S: 91,8%, E: 100% (lymphnode); S: 98%, E: 100% (blood)

[66]

Cuiabá

L. infantum

1,14% (DPP– S: NA, E: NA and ELISA– S: NA, E: NA)

[67]

Florianópolis

L. infantum

1,37% (ELISA– S: NA, E: NA and RIFI– S: NA, E: NA)

[68]

Niterói

NA

15,5% (ELISA– S: NA, E: NA and IFAT– S: NA, E: NA)

[69]

Porto Alegre metropolitan area

NA

4% (PCR– S: NA, E: NA)

[70]

Vitória

NA

13% (ELISA– S: NA, E: NA); 6% (RIFI– S: NA, E: NA)

[74]

Piacatu

L. infantum

16,08% (ELISA– S: NA, E: NA)

[75]

Divinópolis

NA

4,63% (ELISA– S: NA, E: NA and IFAT– S: NA, E: NA)

[76]

Governador Valadares

NA

30,2% (IFAT– S: NA, E: NA)

[77]

NA

29% (ELISA– S: NA, E: NA  and RIFI– S: NA, E: NA)

[78]

Campina Grande

NA

8,4% (IFAT– S: NA, E: NA) and 4,3% (ELISA– S: NA, E: NA)

[79]

Santarém

L. infantum

23,3% (ELISA– S: NA, E: NA and rK39– S:NA, E: NA)

[80]

Europe

Bulgaria

Petrich

L. infantum

NA (IFAT– S: NA, E: NA, PCR– S: NA, E: NA)

[136]

Georgia

Tbilisi and Kutaisi

L. donovani

Complex*

20% (rK39– S: NA, E: NA)

[137]

Kvareli and Sagarejo districts 

19.5% and 11.4% (rK39– S: NA, E: NA)

[138]

Germany

NA

NA

11.8% (ELISA– S: NA, E: NA)

[139]

Hungary

Tolna province

L. infantum

30% (IFAT– S: NA, E: NA, PCR– S: NA, E: NA)

[140]

Kosovo

Prizren, Gjakova, Rahovec, and Deçan

L. infantum

18.49% (ELISA– S: 95%, E: 96%)

[141]

The Netherlands

NA

NA

32.4% (ELISA– S: NA, E: NA)

[139]

North Macedonia

Skopje and Prilep

L. infantum

2.5% (PCR– S: NA, E: NA)

[142]

Romania

Vâlcea County

L. infantum

NA (FASTest®LEISH)

[143]

Ramnicu Vâlcea

L. infantum

8.75% (ELISA– S: NA, E: NA);10% (PCR– S: NA, E: NA)

   [164]

Galați

NA

8.33% (ELISA– S: NA, E: NA)

[144]

ArgeÈ™ County

L. infantum

20.1% (PCR– S: NA, E: NA)

[145]

Slovenia

Kostelo

L. infantum

NA (IFAT– S: NA, E: NA)

[146]

Switzerland

NA

NA

12.2% (ELISA– S: NA, E: NA)

[139]

United Kingdom

NA

L. infantum

NA (ELISA– S: NA, E: NA, PCR– S: NA, E: NA) 

[147]

IFAT: Indirect immunofluorescence reaction; ELISA: Enzyme-linked immunosorbent assay; DAT: Direct agglutination test; PCR: Polymerase Chain Reaction. NA: Not applicable; S: Sensibility of the test; E: Specificity of the test. *Classification that encompasses the species L. donovani, L. infantum (Old world) and L. chagasi (New World).

-Line 78. It is also endemic for cutaneous leishmaniasis. Please, reformulate the sentence.

As suggested, we clarified the sentence as follows:

“… In the African continent, the most VL cases occur in northern countries, belonging to the Mediterranean basin and East Africa, where the main species are L. infantum and L. donovani, respectively [18]…”

-Line 85. Is this difference statistically significant?. Please include it.

We accepted the suggestion, and the revised sentence appears as follows:

“… Also in Algiers, Adel et al. [21] showed in a study using 462 dogs between 2004-2005 that stray dogs were those with the highest prevalence of leishmaniasis (11.7%), followed by animals from the national guard (9.7%) and those that lived on farms (5.9%), but there was no statistical difference between these groups…”

-Line 87: please cite the factor involved: outbreak, climate,..Was it statistically significant?. Please include it.

We accepted the suggestion:

“… Between 2008-2009, there was an increase in the number of cases of CVL, from west to east, in the country’s coastal region between the cities of Tlemcen and Jijel, that could be due to the increase in rainfall in the region, with some cities showing a statistical difference in the rise of CVL prevalence during this period [22]…”

-Line 99: Is Medjez El Babhow in the north?, how was measured this infectivity? Please include it.

We have made appropriate changes according to the suggestion ant the revised sentence appears as follows:

“… In the city of Medjez El Bab, Beja Governorate, in northern Tunisia, the canine infection rate in 1994 and 1995 was 18% and 22.3%, respectively, and the positivity rate was based on serological tests considering both tests used as Enzyme-Linked Immunosorbent Assay (ELISA) and immunofluorescence antibody test (IFAT)...”

-Line 103. Indicate the date of both studies for comparison.

The requested information was included in the sentences, as follows:

“…In the Kairouan Governorate, in the central region of the country, in 2005, the prevalence was 21% in dogs from the rural area of El Alaa, in which 50% of the animals were asymptomatic [31]. Similarly, in a study by Zoghlami et al. [32], in 2013, in seven districts of the same governorate, the overall prevalence was 26.7%. …”

-Line 118: where Barbar El Fugarra  is located?, North of the country?, Please include it.

We agree with the Reviewer #1 and the missing information was included in the manuscript as follows:

“… While in Sudanese territory, Dereure et al. [36] were the first to advance the possibility of the dog acting as a reservoir host in Barbar El Fugarra, in the state of Gedaref, eastern Sudan, considered an endemic area for VL. …”

-Line 119: was it a study for estimating the seroprevalence in random dogs or with clinical signs of leishmaniasis?, the rates are very high. This question should be taken into account when describe all the result of the review.

We have included additional information in the sentence to clarifly ths issue. The revised sentence appears in the manuscript as follos:

“… At the same location, a study conducted by Dereure et al. [37] involving dogs with poor body condition observed a seroprevalence for leishmaniasis of 72.5% (37/51), 74.3% (26/35), and 42.9% (15/35) in the years 1998, 1999, and 2000, respectively. ”

-Line 120: where the villages along the Rahad River are located?, West of the country?. Please include it.

As requested, we have included the missing information, as fllows:

“… In contrast, a study by Hassan et al. [38] in 10 villages along the Rahad River, located in eastern Sudan, in 2002 showed that 6.9% of the dogs were serologically positive for L. donovani…”

-Results from study 175 seems to be affected by potential inespecificity of the technique. Similarly, results from study 42 could potentially be inaccurate (never have I see this figure before). It is also extrange the results from study 178. Is there any detail regarding these studies that could be introduced to clarify these outlier results?. Are these studies well designed?

We are thankful for the comments of Reviewer #1. In fact, due to the fact that we found these studies to have certain concerns in their experimental design, we decided to remove them from the article.

-Please include in Table 1 the studies from The Americas. In addition, other studies are missing in that Table. Please complete Table 1 or include all the studies cited in a supplementary table.

As requested, we have added some "The Americas" studies in Table 1 that appears in the revised manuscript as follows:

Table 1. Percentage of CVL positive dogs and Leishmania species described in African, Asian, American and European countries.

Continent

Country

Location

Parasite species

% of dogs positive for Leishmania (diagnostic test used)

References

Africa

Ethiopia

Amhara region

L. donovani Complex*

3.8% (RIFI – S: NA, E: NA ; and ELISA– S: NA, E: NA); 2.8% (PCR– S: NA, E: NA)

[148]

Kafta Humera district

L. donovani

27.7% (DAT– S: NA, E: NA); 14.8% (KDRT– S: NA, E: NA)

[149]

Addis Zemen, Humera, and Sheraro

L. donovani

Complex*

5.9% (PCR– S: NA , E: NA)

[150]

Benishangul-Gumuz region

NA

13.9% (rK39 ICT– S: NA, E: NA); 5.6 (DAT– S: NA, E: NA)

[151]

Burkina-Faso

Bobo-Dioulasso

L. infantum

5.88% (Serological test– S: NA, E: NA)

[39]

Côte d'Ivoire

Abidjan and Yamoussoukro cities

L. infantum

8.9% (PCR – S: , E: or IFAT– S: NA, E: NA)

[40]

Nigeria

Oyo, Ogun, and Kwara

NA

4.4% (ELISA– S: NA, E: NA)

[41]

Zambia

Southern Province

L. infantum

NA (Serological tests – S: NA, E: NA and PCR)

[42]

Asia

Iran

Meshkin-Shahr district

L. infantum

15.8% (DAT– S: NA, E: NA)

[152]

Boyer Ahmad district

L. infantum

10% (DAT– S: NA, E: NA)

[153]

Kouhsar district

L. infantum

3.6% (DAT– S: NA, E: NA)

[154]

Khorasan Razavi province

L. infantum

7.6% (IFAT– S: NA, E: NA)

[155]

Kerman province

L. infantum

11.25% (DAT– S: NA, E: NA)

[156]

Kerman and Sistan-Baluchestan provinces

L. infantum

15.4% (ELISA– S: NA, E: NA)

[157]

Tehran and Alborz provinces

L. infantum

4.9% (DAT– S: NA, E: NA)

[158]

Meshkin-Shahr district

L. infantum

23.4% (DAT– S: NA, E: NA)

[159]

Hamedan province

L. infantum

3.95% (ELISA– S: NA, E: NA)

[160]

Jiroft district

L. infantum

7.9% (DAT– S: NA, E: NA)

[161]

Ardabil, Alborz, and East-Azerbaijan provinces

L. infantum

100% (DAT– S: NA, E: NA, rK39– S: NA, E: NA, and PCR– S: NA, E: NA)

[87]

Alborz province

L. infantum

2.97% (DAT– S: , E:)

[162]

Golestan province

L. infantum

18% (PCR– S: , E:)

[163]

Nepal

Kathmandu

L. donovani

Complex*

18.57% (PCR– S: NA, E: NA)

[96]

Pakistan

Chilas, Abbotabad, Bagh, Poonch, and Muzafarabad districts

L. donovani

Complex*

18% (DAT– S: NA , E: NA); 26.6% (ELISA– S: NA, E: NA)

[97]

Philippines

NA

L. infantum

NA

[101]

Vietnam

NA

L. infantum

NA

[101]

America

Brazil

Belo Horizonte

L. infantum

64,6% (IFAT– S: NA, E: NA)

[63]

L. infantum

56,7 % (PCR– S: NA, E: NA)

[64]

Boa Vista

L. infantum

10,3% (RIFI– S: NA, E: NA)

[65]

Baixada Cuiabana

L. infantum

16,58% (PCR– S: 94,7%, E:100% (bone marrow); S: 91,8%, E: 100% (lymphnode); S: 98%, E: 100% (blood)

[66]

Cuiabá

L. infantum

1,14% (DPP– S: NA, E: NA and ELISA– S: NA, E: NA)

[67]

Florianópolis

L. infantum

1,37% (ELISA– S: NA, E: NA and RIFI– S: NA, E: NA)

[68]

Niterói

NA

15,5% (ELISA– S: NA, E: NA and IFAT– S: NA, E: NA)

[69]

Porto Alegre metropolitan area

NA

4% (PCR– S: NA, E: NA)

[70]

Vitória

NA

13% (ELISA– S: NA, E: NA); 6% (RIFI– S: NA, E: NA)

[74]

Piacatu

L. infantum

16,08% (ELISA– S: NA, E: NA)

[75]

Divinópolis

NA

4,63% (ELISA– S: NA, E: NA and IFAT– S: NA, E: NA)

[76]

Governador Valadares

NA

30,2% (IFAT– S: NA, E: NA)

[77]

NA

29% (ELISA– S: NA, E: NA  and RIFI– S: NA, E: NA)

[78]

Campina Grande

NA

8,4% (IFAT– S: NA, E: NA) and 4,3% (ELISA– S: NA, E: NA)

[79]

Santarém

L. infantum

23,3% (ELISA– S: NA, E: NA and rK39– S:NA, E: NA)

[80]

Europe

Bulgaria

Petrich

L. infantum

NA (IFAT– S: NA, E: NA, PCR– S: NA, E: NA)

[136]

Georgia

Tbilisi and Kutaisi

L. donovani

Complex*

20% (rK39– S: NA, E: NA)

[137]

Kvareli and Sagarejo districts 

19.5% and 11.4% (rK39– S: NA, E: NA)

[138]

Germany

NA

NA

11.8% (ELISA– S: NA, E: NA)

[139]

Hungary

Tolna province

L. infantum

30% (IFAT– S: NA, E: NA, PCR– S: NA, E: NA)

[140]

Kosovo

Prizren, Gjakova, Rahovec, and Deçan

L. infantum

18.49% (ELISA– S: 95%, E: 96%)

[141]

The Netherlands

NA

NA

32.4% (ELISA– S: NA, E: NA)

[139]

North Macedonia

Skopje and Prilep

L. infantum

2.5% (PCR– S: NA, E: NA)

[142]

Romania

Vâlcea County

L. infantum

NA (FASTest®LEISH)

[143]

Ramnicu Vâlcea

L. infantum

8.75% (ELISA– S: NA, E: NA);10% (PCR– S: NA, E: NA)

   [164]

Galați

NA

8.33% (ELISA– S: NA, E: NA)

[144]

ArgeÈ™ County

L. infantum

20.1% (PCR– S: NA, E: NA)

[145]

Slovenia

Kostelo

L. infantum

NA (IFAT– S: NA, E: NA)

[146]

Switzerland

NA

NA

12.2% (ELISA– S: NA, E: NA)

[139]

United Kingdom

NA

L. infantum

NA (ELISA– S: NA, E: NA, PCR– S: NA, E: NA) 

[147]

IFAT: Indirect immunofluorescence reaction; ELISA: Enzyme-linked immunosorbent assay; DAT: Direct agglutination test; PCR: Polymerase Chain Reaction. NA: Not applicable; S: Sensibility of the test; E: Specificity of the test. *Classification that encompasses the species L. donovani, L. infantum (Old world) and L. chagasi (New World).

-Line 131: What technique was used for studies 48 and 49?, Please include it.

-Line 131: was it (48) a study for estimating the seroprevalence in random dogs or with clinical signs of leishmaniasis?, the rates are very high. As previously asked, this question should be taken into account when describe all the result of the review.

We have made appropriate changes in the revised version of the manuscript as requested and the revised text appears as follows:

“… In this study the dogs were diagnosed by serological tests (such as the rK39-immunochromatographic test (rK39-ICT) and IFAT) in addition to molecular testing (nested-PCR). One group came from two shelters, while the other consisted of dog’s owners who visited a local veterinary clinic, including both symptomatic and asymptomatic animals. In Puerto Iguazú, located in the same province, 7.17% of 209 dogs evaluated were considered positive for L. infantum [48], using rK39-ICT, IFAT and nested-PCR techniques.  …”

-Line 141: indicate incidence (highest number of cases). Maybe you should describe difference of incidence and prevalence at some point, introduction?

In that specific case, we were unable to find exact data such as the highest number of cases; the authors described only the most cases recorded during that period and place.

-Line 142 and 144: where Middle Valley region of the Magdalena River is located?, Neiva in the Huila departments?, Please include it.

Neiva it is a city that is in the department of Huila. (Department is like state in Colombia)

We have included the missing information and the revised sentence appears as follows:

“… In Colombia, VL has two sources of transmission: in the northern departments of Sucre, Bolívar, and Córdoba, where the highest number of cases are registered; and the Middle Valley region of the Magdalena River, between the central and eastern parts of the Colombian Andes Mountain range in the departments of Cundinamarca, Tolima, and Huila [54]…”

-Line 146: where Cali is located?, Please include it. It is very difficult to follow the rationale of the prevalence without a map or geographical details.

We have made appropriate changes in the revised text including the missing information:

“… Interestingly, Arbeláez et al. [56] reported the first case of urban CVL in the city of Cali, in the Valle del Cauca department, southwestern Colombia, in a dog that had never lived in an endemic area and only roamed in areas where human or canine cases of the disease had not been reported …”

-Line 148: were no description of dogs infected with L. infantum before?. where is located?, Please include it. It is very difficult to follow the rationale of the paper without a map or geographical details.

We agree with the Reviewer #1 and appropriated changes were done including geographical details as requested:

“… More recently, dogs infected with L. infantum were found within the urban perimeter of Sincelejo. The frequency of dogs with CVL is as high as the frequencies found 8 years ago in the rural areas of this municipality, which is in the Department of Sucre [57], located in the Colombian Caribbean, as well as in Pradera and Florida, in Valle del Cauca, southwest Colombia [58].…”

-Line 159: was it a study for estimating the seroprevalence in random dogs or with clinical signs of leishmaniasis?, the rates are very high. This question should be taken into account when describe all the result of the review.

We are thankful to detailed analysis by Reviewer #1 and the additional information was included in the manuscript as follows:

“… In a study conducted with randomly sampled dogs from February to March 2019 on El Tigre Island, located in Amapala, Honduras, the seroprevalence of L. infantum infection was 41% (44/107), according to the Dual-path Platform (DPP) rapid test and ELISA. Additionally, the presence of parasite DNA was detected in 94% of the dogs that tested positive in serological tests [60]…”

-Line 174: indicate prevalence and if this was autochthonous infection or imported infection.

In this study the authors described the autochthonous infection and the sentence of revised manuscript was adapted with the missing information as follows:

“…In Uruguay an autochthonous outbreak that occurred in Arenitas Blancas, in the department of Salto, 22% of the animals studied tested positive for leishmaniasis [52]…”

-Line 189: where is located?, Please include it. 

As requested, we have included the location as follows:

“… Samples from 189 dogs, obtained in the town of Alfei Menashe, in Shomron region, were analyzed and 3.3% of the animals tested positive for infection by L. infantum [91]…”

-Line 196 and 199: data showed the prevalence and incidence in the past. Current data indicate that India is not anymore in those figures after the kala-azar elimination program. Please correct it.

We decided to remove this part of the text, as the data we had included no longer matched the local reality.

-Line 202. On Oct 31, 2023, WHO announced that Bangladesh has eliminated visceral leishmaniasis (also known as kala-azar) as a public health problem, the first country worldwide to do so. This elimination status was validated by WHO after Bangladesh reported less than 1 cases per 10 000 people in the previous 3 consecutive years. Please correct it.

We decided to remove this part of the text, as the data we had included no longer matched the local reality.

-Line 203: What technique was used for studies 99, 115 and 116?, Please include it. Why are not included in Table 1?

We decided to remove this part of the text, as the data we had included no longer matched the local reality.

-Line 210: indicate the prevalence in China and technique used to measure prevalence.

We have included the missing information as requested:

“...Shang et al. [98] using real-time PCR, were able to detect Leishmania DNA in dogs from Wenchuan (23/98=23,5%), Heishui (20/71=28,2%) and Jiuzhaigou (35/145=24,1%) counties, all located in Sichuan province, southwest China, where the total positivity rate was 24.8%...”

-Line 233: please cite the factor involved: climate,..

As requested, we have included additional information as follows:

“… However, there has been an increase in the population of vectors and dogs, due to the end of malaria and echinococcosis control programs, which directly affected the vector and dog population on the island, resulting in the resurgence of the CVL in several areas in the south [107,108]…”

-Line 253: references are lacking, please include them. Cite the factor involved: climate, ..

We have changed the text including the missing information that appears as follows:

“…This variation in the prevalence of CVL cases in Greek territory is related to the varied geographical, climatic and socioeconomic conditions in the country as well as the existence of dogs and vectors [116]…”

-Line 254: indicate a range of prevalence and if it is affected by geographical distribution.

As requested, we have included additional information, as follows:

“…The main endemic areas for CVL in Italy include the central and southern regions and the islands of Sicily and Sardinia. In a study conducted on the island of Sardinia involving dogs from a local kennel and those attending by veterinary hospital. The authors examined 1,147 and 15.4% (177/1,147) were seropositive dogs, similar with the seroprevalence of 17.7% previously reported for the mainland part of the country [117]…”

-Line 262: why is it important?, geographical area?. What about Italian Alps?

Because San Marino is an autonomous country located in the middle of Italian territory, and this country is endemic for CVL, it results in the entire Italian peninsula also being endemic.

-Line 275: was it a study for estimating the seroprevalence in random dogs or with clinical signs of leishmaniasis?, the rates are very high. This question should be taken into account when describe all the result of the review.

We have clarified this issue in revised manuscript and the text appears as follows:

“…In other study from May 2011 to February 2014, a total of 230 dogs from veterinary medical centers and animal shelters in the southern Portugal were randomly separate in two groups. One analyzed group presented no clinical signs compatible with CVL and the prevalence was 69% (107/155). The other group with clinical signs compatible with CVL the prevalence was 42,7% (32/75) [129]…”

-Line 283: was it a study for estimating the seroprevalence in random dogs or with clinical signs of leishmaniasis?, the rates are very high. This question should be taken into account when describe all the result of the review.

The random analysis information was included in the text as follows:

“…The CVL seroprevalence rate after a random analysis varies from 2% to 57.1% between regions [133]…”

-Line 403: please confirm that 43.4% of dogs, without clinical sign, were positive for PCR. It is a high rate in blood for asymptomatic dogs. If so, was it influenced by some factors?

That’s correct, animals without clinical signs test positive for CVL assessed by qPCR tests. CVL is considered a chronic and immunological disease. Animals can be positive without clinical signs because the imune system can control the disease ongoing especially at initial stage of infection. However, since the dog is highly susceptible to the L. infantum infection (like humans) there is a parasitism progression, and the clinical signs appear. This is one of the challenges in controlling the canine disease.

-Line 458: how much? Significantly?

We have included additional information in the text to clarify this issue and the revised sentence appears as follows:

“…However, dog screening and faster culling intervention (30 days after CVL diagnosis) during two years of VL transmission in the north of Minas Gerais State (Brazil) reduced the number of human cases in 75% and canine disease cases as recently reported [236]...”

-In my opinion, paragraph 5 only described elimination of infected dogs. I miss information about vaccination and collars, important tools for controlling leishmaniasis. Introduce please some studies about vaccination and collars, or change the title of the paragraph.

We agree with the Reviewer #1 and adapted the title according to the major goal of this Section, as follows:

“5. The euthanasia challenges for controlling the spread of canine visceral leishmaniasis”

-Line 462: humans are also a source of infection for sandflies, please correct.

We changed the sentence that appears as follows:

 “… The dog is considered a key element in the VL cycle caused by L. infantum, especially in the Americas and the Mediterranean basin.”

-Line 476: It seems reasonable to comment here some of the outliers described previously, example study 178 among others.

Due to the fact that we found these studies to have certain concerns in their experimental design, we decided to remove them from the article.

-Line 493: Brazil is the only country cited. Other general measures from other countries should be included.

Does any country have a national plan including dog control?.

Countries in South America have national plans to control dogs after testng positive for Visceral Leishmaniasis. In countries as Africa and Asia, the control measures do not include dogs, and in Europe, the use of repellents and vaccination is recommended.

We have made appropriated changes in the revised manuscript and the text appears as follows:

“… Countries such as Argentina [250], Brazil [251] , Colombia [252], Paraguay [253], Uruguay [254], among others in South America, recommend the euthanasia of L. infantum-infected dogs…”

-Line 501: indicate levels of induced protection and differentiate them from capability to infect.

We accepted the Reviewer #1 suggestion and the revised text appears as follows:

“…The effectiveness of vaccines currently on the European market varies. CaniLeish® prevents clinical signs in 68.4% of cases and provides a protection level—defined as the percentage of vaccinated animals that remain asymptomatic (and not in terms of parasite protection)—of 92.7%. Similarly, LetiFend® showed a 72% effectiveness in preventing clinical signs and helps reduce these signs in infected animals [257,258]. The levels of protection provided by those vaccines are not sufficient to prevent infection by L. infantum [17,256,259]. In fact, as shown in a study carried out by Cotrina et al. [258], in which dogs vaccinated with LetiFend® showed 9.4% positivity for L. infantum, with no statistical difference between the control group (16.1%). For CaniLeish®, Oliva et al. [257] through a randomized double-blind controlled trial, in an endemic area for CVL, showed that vaccinated dogs presented 50.22% positivity for Leishmania, while the control group presented 66.66%....”

-Line 508: indicate levels of effectiveness.

We have included the levels of effectiveness as requested:

“…In a mathematical model, Sevá et al. [220] found that applying collars to 90% of dogs effectively eliminated the disease among humans and nearly wiped out the infection in the dog population as well…”

-Line 509: from where?, like in line 493 only Brazil?

Yes, it´s a strategy used by the Brazilian Ministry of health, but that strategy is also used in Europe and Iran. We have changed the text that appears in revised manuscript as follows:

“…. Nonetheless, these alternatives have been used together with repellent collars, which have been widely accepted by dog owners in Europe [171] and are highly regarded for their effectiveness, according to studies by Gavgani et al. (Iran) [190], Sevá et al. (Brazil) [220] and, Ribas et al. (Brazil) [262] … In 2021, the Brazilian Ministry of Health made the use of collars official as a way of controlling the disease in municipalities where transmission varies from high to very intense [263]…”

-Line 514: new technologies?, give examples.

In this case is not development of new technologies but it is improvement of new immunobiologicals to block CVL transmission. This sentence was corrected and addition al information was included as follows:

“…Improvements in technology that facilitate the development of vaccines capable of disrupting the transmission of L. infantum from infected dogs could serve as a highly effective measure in preventing its spread. In fact, some studies have been described the sandfly antigens to interfere in the vector life cycle in addition to control de L. infantum infection after blood meal that is compatible with de hypothesis to effective control of CVL transmission [17,256,259,261,264,265]…

-Line 518: Is including here anthroponotic countries?, Although there might be transmission caused by the dogs in these countries, do you conclude that the dog is the main actor there?

In those anthroponotic countries the dogs is not the main actor. For this reason, the text was adapted as follows:

“From the insights provided in this review, it becomes evident that dogs play a pivotal role in the transmission cycle of visceral leishmaniasis, particularly in regions where the disease is considered zoonotic and where instances of canine visceral leishmaniasis have been on the rise…”

-Line 521: is canine visceral leishmaniasis being spread due to poverty?

In fact, socioeconomic factors could be a cause to spread the CVL. We have adapted the text that appears in the revised sentence as follows:

“…Based on the information presented in this review, it becomes evident that dogs play a pivotal role in the transmission cycle of visceral leishmaniasis caused by L. infantum. However, it's crucial to acknowledge that various other factors contribute significantly to this transmission dynamic. Environmental conditions, socioeconomic factors, climatic variations, as well as shortcomings in disease treatment and control efforts all play significant roles in shaping the prevalence and spread of the disease…

Reviewer 2 Report

Comments and Suggestions for Authors

Dear Authors I reviewed your manuscript entitled "Global distribution of canine visceral leishmaniasis and the role of the dog in the epidemiology of the disease " and can conclude that it can not be accepted at all. Here are some of my concerns:

1)      there are too many references. If you compare number of pages covering references – it is half of the manuscript-way to much.

2)      Wrong citations: After checking the contents of the references – there are to many wrong citations. First citation (1) -wrong; second citation (2-5)-wrong; third citation (6)-wrong etc. you can find the others in corrected manuscript file.

3)      Please cite the original references and not the citation of the citation. Write only data which you can support by references and then you’ll be able to avoid wrong citations. Be more carefull

4)      Title has to be changed in “Global distribution of canine leishmaniosis and the role of the dog in the epidemiology of the disease. “ There is no canine visceral leishmaniasis – symptoms of the disease are cutaneous, mucocutaneous and visceral – so it should be written just “canine leishmaniosis”.

5)      you should arrange the references according to the Instructions for the authors of the Journal

Best regards

Comments on the Quality of English Language

English language needs minor editing

Author Response

Reviewer #2:

Comments and Suggestions for Authors

Dear Authors I reviewed your manuscript entitled "Global distribution of canine visceral leishmaniasis and the role of the dog in the epidemiology of the disease " and can conclude that it can not be accepted at all. Here are some of my concerns:

1) there are too many references. If you compare number of pages covering references – it is half of the manuscript-way to much.

2) Wrong citations: After checking the contents of the references – there are to many wrong citations. First citation (1) -wrong; second citation (2-5)-wrong; third citation (6)-wrong etc. you can find the others in corrected manuscript file.

3) Please cite the original references and not the citation of the citation. Write only data which you can support by references and then you’ll be able to avoid wrong citations. Be more carefull

4) Title has to be changed in “Global distribution of canine leishmaniosis and the role of the dog in the epidemiology of the disease. “ There is no canine visceral leishmaniasis – symptoms of the disease are cutaneous, mucocutaneous and visceral – so it should be written just “canine leishmaniosis”.

5) you should arrange the references according to the Instructions for the authors of the Journal

We are thankful for the comments of the Reviewer #3. We considered those comments as opinion based on his(er) experience. An example is the terminology “canine visceral leishmaniasis” widely used in the scientific community that used as keyword in the pubmed resulted in 2,732 publications (please see the link: https://pubmed.ncbi.nlm.nih.gov/?term=canine+visceral+leishmaniasis&sort=pubdate&size=200 ). Furthermore, we have worked throughout the manuscript to correct all the missing information and after all the suggestion from the Reviewers #1-3, we believe that our manuscript is now with the high quality compatible with published papers in the Pathogens Journal.

Reviewer 3 Report

Comments and Suggestions for Authors

This review of canine visceral leishmaniasis aims to provide an overview of the distribution of the disease in endemic areas of the world, to show the role of dogs in the transmission of this disease and the measures taken to control the disease in certain regions of the world.

This review details the epidemiology in the various endemic areas, displays the studies that have shown the reservoir role of symptomatic and asymptomatic dogs, as well as the strategies that have attempted to control the spread of this disease.

Minor comments:

Lines 47-48, perhaps prefer the phrasing: "12 million infected people" rather than "12 infected million people".

Perhaps provide percentages to maintain uniformity: e.g. lines 94-95, 156-157.

Line 109-111, perhaps indicate that this study concerns asymptomatic dogs.

Line 131, it would appear to be 57.3% instead of 58.3%

Line 44, replace Flanders with Flandes.

Line 150-156, possibly indicate that French Guiana is not an endemic area for VL.

Line 158, indicate the full name of DDP.

Remove one of the repetitions on lines 291 and 292 "for the first time".

Author Response

Reviewer #3

Comments and Suggestions for Authors

This review of canine visceral leishmaniasis aims to provide an overview of the distribution of the disease in endemic areas of the world, to show the role of dogs in the transmission of this disease and the measures taken to control the disease in certain regions of the world. This review details the epidemiology in the various endemic areas, displays the studies that have shown the reservoir role of symptomatic and asymptomatic dogs, as well as the strategies that have attempted to control the spread of this disease.

 We acknowledge the Reviewer #3 for the comment, which demonstrate that our major message was well accepted.

Minor comments:

Lines 47-48, perhaps prefer the phrasing: "12 million infected people" rather than "12 infected million people".

We agree with the Reviewer suggestion ant the sentence was modified as follows:

“… Worldwide, it is estimated that there are about 12 million infected people, with approximately 700,000 to 1 million new cases reported each year [2–5]…”

Perhaps provide percentages to maintain uniformity: e.g. lines 94-95, 156-157.

We acknowledge the suggestion and in the revised manuscript the text appears as follows:

“… Recently, 29 dogs infected with L. infantum have been found in northwestern Morocco from such urban centers as Rabat (21/29=72,4%), Benslimane (2/29=6,9%), Casablanca (2/29=6,9%), Kenitra (1/29=3,45%), Fes (1/29=3,45%), Rommani (1/29=3,45%) and Jorf al Melha (1/29=3,45%)[27]…”

“…In a study conducted in Venezuela, the seroprevalence of dogs infected with L. in-fantum was found to be 5.2% (3/58) in the state of Lara. In the state of Yaracuy, no sero-positive dogs for L. infantum's antigen were detected. However, qPCR testing identified one positive case [59]…”

Line 109-111, perhaps indicate that this study concerns asymptomatic dogs.

We accepted the suggestion and the revised text appears as follows:

“…In Egypt, asymptomatic infected dogs were found in five provinces in the north of the country, with a seroprevalence rate of 21.3%, with the highest being found in the provinces of Giza and Cairo [34]…”

Line 131, it would appear to be 57.3% instead of 58.3%

We accepted the suggestion and the revised sentence appears as:

“…In Argentina, a country where the number of human VL cases rose in 2019, presenting nine cases (0.35% of cases in the Americans) [44], infected dogs were found in the city of Posadas, province of Misiones [45,46], where Cruz et al. [47] obtained 57.3% infection rate of dogs by L. infantum (63/110). …”

Line 144, replace Flanders with Flandes.

We apologize for this. We corrected the sentence that appears as follows:

“… Picón et al. [55] found a 12% seroprevalence in 2016-2017 when analyzing dogs in the cities of Guamo, Ortega, Flandes, Coyaima, and Melgar in Tolima department and Neiva in the Huila department …”

Line 150-156, possibly indicate that French Guiana is not an endemic area for VL.

We agree with this sentence, French Guiana is not an endemic area for VL, so we relocated that paragraph at the end of the new topic “The Americas” for a better understanding, as follows:\

“… It is known that the disease has reached areas where it had previously not been present. The expansion of CVL to non-endemic regions can be illustrated by reports, such as the occurred in the French overseas department, French Guiana, dogs with autochthonous infection by L. infantum were reported in 2016 in Cayenne through molecular tests, with a prevalence of 3.1%. In addition, it was observed in dogs that performed military work, which had negative molecular and serological results prior to going to French Guiana, yet tested positive for infection when they returned to France after four months in the region, suggesting the possible existence of a autochthonous cycle of VL transmission in the department [82]… “

Line 158, indicate the full name of DDP.

We agree with the suggestion and the sentence was rephased and appears as follows:

 “… In a study conducted with randomly sampled dogs from February to March 2019 on El Tigre Island, located in Amapala, Honduras, the seroprevalence of L. infantum infection was 41% (44/107), according to the Dual-path Platform (DPP) rapid test and ELISA...”

Remove one of the repetitions on lines 291 and 292 "for the first time".

We agree with the suggestion and the sentence was corrected as follows:

 “…In addition, for the first time, the seroprevalence of L. infantum on the Canary Islands was reported with 2.45% of dogs testing also positive for leishmaniasis [132]...”

Round 2

Reviewer 1 Report

Comments and Suggestions for Authors

Authors have positively answered to all the suggested corrections and now the manuscript is ready to be published.

Minor comments:

-In Table 1, I suggested to include Sensibility and Specificity for the detailed studies. As described by authors, most of the studies did not include this, consequently the table has a quite high number of NA, and it does not look good or easy to read. Please, remove the Sensibility and Specificity from the table and just add a sentence into the text (out of the table) describing the Sensibility and Specificity only of the studies that included it (65, 66, 141). Somenthing like: Regretably most of the studies do not include the Sensibility and Specificity of the tests used for CVL diagnosis/detection, with the exception of XX [PCR– S: 94,7%, E:100% (bone marrow)]; XX [S: 91,8%, E: 100% (lymphnode)]; S: 98%, E: 100% (blood) , XX [ELISA- S: 95%, E: 96% (sera)].

-Line 553: L.infantum should be in italics.

Author Response

Reviewer #1:

Comments and Suggestions for Authors

Authors have positively answered to all the suggested corrections and now the manuscript is ready to be published.

Minor comments:

-In Table 1, I suggested to include Sensibility and Specificity for the detailed studies. As described by authors, most of the studies did not include this, consequently the table has a quite high number of NA, and it does not look good or easy to read. Please, remove the Sensibility and Specificity from the table and just add a sentence into the text (out of the table) describing the Sensibility and Specificity only of the studies that included it (65, 66, 141). Somenthing like: Regretably most of the studies do not include the Sensibility and Specificity of the tests used for CVL diagnosis/detection, with the exception of XX [PCR– S: 94,7%, E:100% (bone marrow)]; XX [S: 91,8%, E: 100% (lymphnode)]; S: 98%, E: 100% (blood), XX [ELISA- S: 95%, E: 96% (sera)].

We acknowledge the Reviewer #1 for the critical reading of the manuscript. We agree with the suggestion, and the revised sentence was clarified as follows:

Continent

Country

Location

Parasite species

% of dogs positive for Leishmania (diagnostic test used)

References

Africa

Ethiopia

Amhara region

L. donovani Complex*

3.8% (RIFI and ELISA); 2.8% (PCR)

[148]

Kafta Humera district

L. donovani

27.7% (DAT); 14.8% (KDRT)

[149]

Addis Zemen, Humera, and Sheraro

L. donovani

Complex*

5.9% (PCR)

[150]

Benishangul-Gumuz region

NA

13.9% (rK39 ICT); 5.6 (DAT)

[151]

Burkina-Faso

Bobo-Dioulasso

L. infantum

5.88% (Serological test)

[39]

Côte d'Ivoire

Abidjan and Yamoussoukro cities

L. infantum

8.9% (PCR or IFAT)

[40]

Nigeria

Oyo, Ogun, and Kwara

NA

4.4% (ELISA)

[41]

Zambia

Southern Province

L. infantum

NA (Serological tests and PCR)

[42]

Asia

Iran

Meshkin-Shahr district

L. infantum

15.8% (DAT)

[152]

Boyer Ahmad district

L. infantum

10% (DAT)

[153]

Kouhsar district

L. infantum

3.6% (DAT)

[154]

Khorasan Razavi province

L. infantum

7.6% (IFAT)

[155]

Kerman province

L. infantum

11.25% (DAT)

[156]

Kerman and Sistan-Baluchestan provinces

L. infantum

15.4% (ELISA)

[157]

Tehran and Alborz provinces

L. infantum

4.9% (DAT)

[158]

Meshkin-Shahr district

L. infantum

23.4% (DAT)

[159]

Hamedan province

L. infantum

3.95% (ELISA)

[160]

Jiroft district

L. infantum

7.9% (DAT)

[161]

Ardabil, Alborz, and East-Azerbaijan provinces

L. infantum

100% (DAT, rK39, and PCR)

[87]

Alborz province

L. infantum

2.97% (DAT)

[162]

Golestan province

L. infantum

18% (PCR

[163]

Nepal

Kathmandu

L. donovani

Complex*

18.57% (PCR)

[96]

Pakistan

Chilas, Abbotabad, Bagh, Poonch, and Muzafarabad districts

L. donovani

Complex*

18% (DAT); 26.6% (ELISA)

[97]

Philippines

NA

L. infantum

NA

[101]

Vietnam

NA

L. infantum

NA

[101]

America

Brazil

Belo Horizonte

L. infantum

64,6% (IFAT)

[63]

L. infantum

56,7 % (PCR)

[64]

Boa Vista

L. infantum

10,3% (RIFI)

[65]

Baixada Cuiabana

L. infantum

16,58% (PCR)

[66]

Cuiabá

L. infantum

1,14% (DPP and ELISA)

[67]

Florianópolis

L. infantum

1,37% (ELISA and RIFI)

[68]

Niterói

NA

15,5% (ELISA and IFAT)

[69]

Porto Alegre metropolitan area

NA

4% (PCR)

[70]

Vitória

NA

13% (ELISA); 6% (RIFI)

[74]

Piacatu

L. infantum

16,08% (ELISA)

[75]

Divinópolis

NA

4,63% (ELISA and IFAT)

[76]

Governador Valadares

NA

30,2% (IFAT)

[77]

NA

29% (ELISA and RIFI)

[78]

Campina Grande

NA

8,4% (IFAT) and 4,3% (ELISA)

[79]

Santarém

L. infantum

23.3% (ELISA and rK39)

[80]

Europe

Bulgaria

Petrich

L. infantum

NA (IFAT and PCR)

[136]

Georgia

Tbilisi and Kutaisi

L. donovani

Complex*

20% (rK39)

[137]

Kvareli and Sagarejo districts 

19.5% and 11.4% (rK39)

[138]

Germany

NA

NA

11.8% (ELISA)

[139]

Hungary

Tolna province

L. infantum

30% (IFAT and PCR)

[140]

Kosovo

Prizren, Gjakova, Rahovec, and Deçan

L. infantum

18.49% (ELISA)

[141]

The Netherlands

NA

NA

32.4% (ELISA)

[139]

North Macedonia

Skopje and Prilep

L. infantum

2.5% (PCR)

[142]

Romania

Vâlcea County

L. infantum

NA (FASTest®LEISH)

[143]

Ramnicu Vâlcea

L. infantum

8.75% (ELISA);10% (PCR)

   [164]

Galați

NA

8.33% (ELISA)

[144]

ArgeÈ™ County

L. infantum

20.1% (PCR)

[145]

Slovenia

Kostelo

L. infantum

NA (IFAT)

[146]

Switzerland

NA

NA

12.2% (ELISA)

[139]

United Kingdom

NA

L. infantum

NA (ELISA and PCR) 

[147]

IFAT: Indirect immunofluorescence reaction; ELISA: Enzyme-linked immunosorbent assay; DAT: Direct agglutination test; PCR: Polymerase Chain Reaction. NA: Not applicable. *Classification that encompasses the species L. donovani, L. infantum (Old world) and L. chagasi (New World). Unfortunately, the majority of studies do not provide the sensitivity and specificity of the tests used for CVL diagnosis/detection. Exceptions include: researches 66 [PCR – sensitivity (S): 94.7%, specificity (E): 100% (bone marrow); S: 91,8%, E:100% (lymphnode); S:98%, E:100% (blood)] and 141 [ELISA – S: 95%, E: 96%].

-Line 553: L.infantum should be in italics.

We are thankful for this comment. We have made appropriate changes in the sentence that appears in revised version of the manuscript as follows:

“… cycle in addition to control de L. infantum infection after blood …’

We are grateful for the corrections made to the manuscript, and we believe that the changes have improved its quality. We hope that this revised version is now compatible with the high-quality manuscripts published in Pathogens.

Yours sincerely,

The authors.

Reviewer 2 Report

Comments and Suggestions for Authors

Dear Authors, as you didn't accept my suggestions, your manuscript can0t me accepted.

Best regards

Author Response

We respect the opinion of the reviewer but all the authors considered that all the issues raised during the revision process were addressed according to the suggestions.